# Catalase Activity of IgGs of Patients Infected with SARS-CoV-2

**DOI:** 10.3390/ijms241210081

**Published:** 2023-06-13

**Authors:** Anna S. Tolmacheva, Margarita K. Onvumere, Sergey E. Sedykh, Anna M. Timofeeva, Georgy A. Nevinsky

**Affiliations:** Institute of Chemical Biology and Fundamental Medicine, SB of the RAS, 630090 Novosibirsk, Russia; anny@mail.ru (A.S.T.); m.onvumere@g.nsu.ru (M.K.O.); bezukaf@mail.ru (A.M.T.)

**Keywords:** SARS-CoV-2, COVID-19, catalase activity of IgGs

## Abstract

Coronavirus disease (COVID-19), caused by the SARS-CoV-2 coronavirus, leads to various manifestations of the post-COVID syndrome, including diabetes, heart and kidney disease, thrombosis, neurological and autoimmune diseases and, therefore, remains, so far, a significant public health problem. In addition, SARS-CoV-2 infection can lead to the hyperproduction of reactive oxygen species (ROS), causing adverse effects on oxygen transfer efficiency, iron homeostasis, and erythrocytes deformation, contributing to thrombus formation. In this work, the relative catalase activity of the serum IgGs of patients recovered from COVID-19, healthy volunteers vaccinated with Sputnik V, vaccinated with Sputnik V after recovering from COVID-19, and conditionally healthy donors were analyzed for the first time. Previous reports show that along with canonical antioxidant enzymes, the antibodies of mammals with superoxide dismutase, peroxidase, and catalase activities are involved in controlling reactive oxygen species levels. We here show that the IgGs from patients who recovered from COVID-19 had the highest catalase activity, and this was statistically significantly higher each compared to the healthy donors (1.9-fold), healthy volunteers vaccinated with Sputnik V (1.4-fold), and patients vaccinated after recovering from COVID-19 (2.1-fold). These data indicate that COVID-19 infection may stimulate the production of antibodies that degrade hydrogen peroxide, which is harmful at elevated concentrations.

## 1. Introduction

Coronavirus disease (COVID-19), caused by the SARS-CoV-2 coronavirus, remains a major public health problem. As of 2 April 2023, more than 762 million cases of infection and 6.8 million deaths due to SARS-CoV-2 infection have been recorded [1]. Since the beginning of the pandemic, the pathophysiology of COVID-19 has been extensively studied, and many diagnostic, treatment, and preventive measures have been developed, but there are still questions that require further research. The causes of various post-COVID conditions, including diabetes [2], heart and kidney disease [3,4], thrombosis [5], neurological and autoimmune diseases [6,7], as well as cases of complications after COVID-19 vaccination, such as central nervous system inflammation [8], hyperviscosity syndrome [9], and pericarditis and myocarditis [10,11], are still poorly understood.

It has now been shown that the redox imbalance in COVID-19 can directly affect the spread of viral particles. For example, a certain disulfide–thiol balance of the extracellular medium is required at both stages of virus entry [12]. The reduction in disulfide bonds to thiol groups in angiotensin-converting enzyme 2 (ACE2) and SARS-CoV-2 receptor binding domain (RBD) significantly reduces their binding affinity for each other [13]. Reactive oxygen species (ROS) can oxidize cysteine residues, keeping them in the disulfide form, which can increase the SARS-CoV-2 affinity to ACE2 [14]. On the other hand, SARS-CoV-2 infection directly affects the production of ROS. ACE2 is a central negative regulator of the renin–angiotensin–aldosterone system. The protein acts as a zinc metallopeptidase and activates angiotensin (Ang), converting Ang II to Ang-(1-7) [15]. The binding of the viral S-protein to ACE2 increases the concentration of Ang II, because ACE2 can no longer convert Ang II to Ang 1-7. Under these conditions, Ang II binds to the angiotensin type 1 receptor (AT1R), which stimulates the activity of NADPH oxidase and, thus, increases the production of superoxide anion radicals [16].

Some pathophysiological processes that occur with COVID-19 are also caused by the excessive generation of ROS. SARS-CoV-2 infection can lead to neutrophil infiltration in the pulmonary capillaries and neutrophilia-induced excessive ROS release, which causes damage to alveolar epithelial and endothelial cells and also perpetuates neutrophil activation by initiating a chain reaction [17,18]. Hyperproduction of ROS causes the destruction of erythrocyte membranes, which negatively affects oxygen transfer efficiency, iron homeostasis (Fe^2+^/Fe^3+^ balance), and erythrocyte deformability, contributing to thrombosis [16,19]. The detrimental effects of ROS on pulmonary cell and erythrocyte functions are thought to be the major contributors to acute hypoxic respiratory failure [16]. Excessive formation of ROS also disrupts signaling pathways that affect blood pressure, leading to hypertension [20]. The overexpression of NADPH oxidase 2 (NOX2) is also observed in hospitalized patients with COVID-19 [21].

SARS-CoV-2 infection has been shown to cause a decrease in the body’s antioxidant defenses. It is known that exposure to prooxidants typically leads to nuclear translocation of the nuclear factor erythroid-related factor 2 (Nrf2), which activates antioxidant protection. However, respiratory viral infections inhibit Nrf2-dependent signaling pathways, which, in turn, may contribute to the achievement of inflammation [16,22]. In addition, a decrease in glutathione, peroxiredoxins, and superoxide dismutase (SOD) levels and a reduction in SOD3 and glutathione peroxidase (GPx)-4 gene expression are observed during SARS-CoV-2 infection [21]. Thus, an excess of ROS during COVID-19 causes local and systemic damage to the body.

Catalytically active antibodies (abzymes) hydrolyzing DNA, RNA, peptides, and various proteins, including histones and oligosaccharides, are a specific feature of some autoimmune and viral diseases (see [23,24] for a review). They have been found in patients with systemic lupus erythematosus (SLE), multiple sclerosis (MS), schizophrenia, autoimmune Hashimoto’s thyroiditis, HIV-infected patients, tick-borne encephalitis [25,26], and other diseases [27]. With rare exceptions, such abzymes are absent in the blood of conditionally healthy donors. At the same time, abzymes with redox functions have been found in the blood of all healthy people and animals [28,29,30].

Along with canonical antioxidant defense enzymes, antibodies with similar redox activities, such as superoxide dismutase (SOD) [29,30], H_2_O_2_-dependent peroxidase, and peroxide-independent oxidoreductase activity [31,32], may be involved in the regulation of ROS levels. Such antibodies—abzymes—have been found in healthy individuals, as well as in patients with autoimmune [33,34], neurodegenerative, and infectious diseases [35] (for reviews, see [33,34,35,36,37,38,39,40,41]). Abzymes with peroxidase and oxidoreductase activities were shown to be significantly increased in patients with SLE and MS [34]. It has also been demonstrated that the IgGs and IgAs of patients with acute reactive arthritis [36], early arthritis [37], and breast neoplasms [38] have catalase activity.

Over the last few years, the catalase activity of polyclonal immunoglobulins in the blood of patients with various diseases has been actively studied. IgG catalase activity has been found to be significantly higher in patients with schizophrenia [39] and multiple sclerosis [30] than in apparently healthy donors. A similar situation was revealed in the case of C57BL/6 mice prone to experimental autoimmune encephalitis (EAE) [40]. Catalase activity increases relatively slowly during the spontaneous development of EAE but increases sharply with the acceleration of the disease after immunization of mice with myelin oligodendrocyte glycoprotein or histone–DNA complex [41]. The abzymes of animals, conditionally healthy people, and patients with autoimmune diseases can protect the body from oxidative stress by oxidizing different toxic compounds.

It was interesting to see how the change in catalase activity occurred in patients with COVID-19 and in patients immunized with Sputnik V compared to conditionally healthy donors. In this work, for the first time, an analysis of the catalase activity of these groups was carried out. The analysis of the possible role of abzymes with catalase activity in patients with COVID-19 can improve our understanding of the causes of the various complications of this disease and contribute to the search for therapeutic approaches to prevent its severe course.

## 2. Results

### 2.1. Sample Collection and Characteristics of the Donors

Blood plasma samples were obtained from patients who had recovered from COVID-19 and those who were vaccinated with Sputnik V before and after COVID-19, and there was a control group of conditionally healthy donors. Blood samples from healthy donors were collected before the start of the coronavirus pandemic. The blood of patients who recovered from COVID-19 vaccinated with Sputnik V before and after COVID-19 was collected between October 2020 and May 2021, during the first wave of coronavirus infections (Wuhan variant). The patients were selected based on the results of a questionnaire and an enzyme-linked immunosorbent assay (ELISA) of blood plasma antibodies to the S- and N-proteins of SARS-CoV-2. The study patients and donors had no history of autoimmune diseases and chronic infections. Since COVID-19 infection elicits the production of antibodies to all SARS-CoV-2 proteins, and Sputnik V vaccination produces antibodies only to the S-protein [42], as confirmed by the results of an ELISA, vaccinated patients who did not have COVID-19 were separated as a dependent group from patients vaccinated after contracting COVID-19. As a result, the blood plasma of 28 patients who had recovered from COVID-19 (group 1), 26 patients vaccinated with Sputnik V who had no previous history of COVID-19 (group 2), 13 patients vaccinated after previously recovering from COVID-19 (group 3), and 43 conditionally healthy donors (group 4) were selected for further research. The characteristics of the study groups of patients are presented in Table 1.

### 2.2. IgG Isolation

IgG preparations were isolated by affinity chromatography of blood plasma proteins on Protein G Sepharose under conditions of the removal of proteins not specifically bound to the sorbent and the selective elution of IgGs with a low pH buffer of 2.6, as shown in [43]. This method has previously been shown to obtain homogeneous IgG preparations without contamination from other blood proteins or enzymes [26,44]. Nonspecifically adsorbed proteins and blood plasma lipids were eluted from the sorbent with a buffer containing 1% non-ionic detergent Triton X-100. The specific elution of antibodies was carried out using the acidic buffer. Immediately after elution from the column, the IgG fractions were neutralized and subjected to dialysis. Figure 1 shows a typical chromatographic profile of the plasma components corresponding to one of the donors recovered from COVID-19.

The homogeneity of the IgG preparations was analyzed with SDS-PAGE followed by staining with Coomassie Blue R-250 under typical reducing conditions in the presence of dithiothreitol (DTT) and nonreducing conditions in the absence of DTT. As an example, Figure 2 shows data from the analysis of four IgG_mix_ preparations corresponding to equimolar mixtures of all IgGs corresponding to three groups of patients (1–3, see above) and one group of healthy donors, as well as the Abs of one individual patient recovered from COVID-19 and one preparation of an apparently healthy donor.

Under nonreducing conditions, one protein band was observed corresponding to intact ~150 kDa IgGs and two protein bands with molecular weights of approximately 25 and 50 kDa, corresponding to the light and heavy chains of the Abs after reduction. All IgG preparations were shown to be electrophoretically homogeneous.

To prove that the catalase activity of the IgG preparations belonged to antibodies and was not due to enzyme impurities, we applied several previously developed strict criteria to the activity to ensure it belonged to antibodies [25,26,27,28,39,44]. First, after the gel filtration of the COVID-19-IgG_mix_ (equimolar mixture of 21 preparations) under conditions of the dissociation of noncovalent complexes in an acidic buffer (pH 2.6), the peak catalase activity of the IgGs precisely coincided with the peak (A_280_) of the IgG preparation absorption (Figure 3A). Secondly, the COVID-19-IgG_mix_ preparation was separated with SDS-PAGE (Figure 3B); then, the proteins were extracted from the gel fragments, and the catalase activity of the fractions was determined. It was shown that the catalase activity was recorded only in the regions of the gel corresponding to the position of intact IgGs (Figure 3C). The association of catalase activity with antibodies from healthy donors was previously confirmed [39]. Thus, it was shown that the IgG preparations have catalase activity and do not contain impurities of proteins with this activity.

### 2.3. Catalase Activity of the IgGs

The catalase activity of the IgGs was analyzed using a spectrophotometric method according to [45] in which a decrease in the optical density (A_240_) at λ = 240 nm, caused by the decomposition of hydrogen peroxide, was recorded. Preliminarily, specific concentrations of the IgG preparations were selected, at which a linear dependence of the reaction rate of the hydrogen peroxide decomposition on the concentration of antibodies in the reaction mixture and the incubation time was observed. Therefore, to assess the catalase activity, the values of the apparent *k*cat were used, which were calculated using the formula *k*cat = V (M/min)/(IgG) (M), where V is the rate of H_2_O_2_ decomposition, and (IgG) is the total concentration of IgGs in the reaction mixture. Figure 4 shows typical kinetic curves of the changes in the H_2_O_2_ concentration in the presence of the IgG preparations compared to a control mixture containing no Abs.

Figure 5 shows a comparison of the IgG catalase activity of the four studied groups. All IgG preparations had catalase activity. The median values of the apparent *k*cat of the reaction of hydrogen peroxide decomposition by antibodies for the studied groups of patients and healthy donors decreased in the following order (min^−1^): patients recovered from COVID-19 (group 1)—1.7 × 10^3^; previously not ill patients vaccinated with Sputnik V (group 2)—1.2 × 10^3^; healthy donors (group 4)—0.9 × 10^3^; and patients vaccinated after the disease (group 3)—0.8 × 10^3^. The level of catalase activity of the IgG preparations of the patients who recovered from COVID-19 differed statistically (*p* < 0.05) from the activity of antibodies from the other three groups. At the same time, no statistically significant differences were found between the catalase activity of antibodies of vaccinated patients and healthy donors. This indicates that not all anti-COVID-19 IgGs had catalase activity, and necessitate IgG subfractioning.

### 2.4. Isolation and Characterization of the IgG Subclasses

The IgG subfractions containing two kappa light chains (κ,κ-IgGs) and two lambda light chains (λ,λ-IgGs), as well as bispecific chimeric antibodies with one λ and one κ light chain (κ,λ-IgGs), were isolated with affinity chromatography from an equimolar mixture of the IgG preparations. Columns with KappaSelect and LambdaSelect sorbents containing immobilized special recombinant proteins were used. Two affinity chromatographies were performed, first with KappaSelect and then with LambdaSelect, and vice versa. As an example, Figure 6 shows a scheme for the isolation of κ,κ-IgGs, λ,λ-IgGs, and κ,λ-IgGs from polyclonal electrophoretically homogeneous IgG preparations from healthy donors.

In the first step (Figure 6A), κ,κ-IgG and κ,λ-IgG fractions were obtained using KappaSelect sorbent. The column was washed with special buffers to remove unbound IgGs. Antibodies containing κ light chains were eluted with pH 2.6 buffer. At the second stage (Figure 6B), the κ,κ-IgG and κ,λ-IgG fractions were separated using a LambdaSelect column; κ,κ-IgGs were eluted upon loading, while κ,λ-IgGs were eluted with acidic buffer. Similarly, polyclonal IgGs were loaded onto a LambdaSelect, λ,λ-IgGs and κ,λ-IgGs containing lambda chains were isolated, a κ,λ-IgG preparation was loaded onto a KappaSelect, and, finally, λ,λ-IgGs and κ,λ-IgGs were obtained (Figure 6D). Such κ,κ-IgG, λ,λ-IgG, and κ,λ-IgG preparations were obtained for all four groups.

The relative content of the different subclasses of IgGs (IgG1—IgG4) in κ,κ-IgGs, λ,λ-IgGs, and κ,λ-IgGs was determined with enzyme-linked immunosorbent assays (ELISAs) using total preparations of these IgGs (Table 2).

In the κ,κ-IgG and λ,λ-IgG subfractions corresponding to all studied groups, IgG1 is significantly predominant—the percentage of IgGs of the different subclasses decreases in all IgG preparations in the following order (% of all IgGs): IgG1 (47.5–84.6), IgG2 (8.0–39.0), IgG3 (3.2–19.2), and IgG4 (1.1–10.9).

On average, for all preparations an increase in the content of IgG1 was observed in the following order (% of all IgGs): κ,κ-IgG (74.9–84.2); λ,λ-IgG (70.0–84.6); and κ,λ-IgG (47.5–63.6). A reduced content of IgG2 was observed in the case of λ,λ-IgG (8.0–11.3; average value: 8.3 ± 2.6%) and was markedly higher in the κ,κ-IgGs (9.3–19.0; average value: 14.2 ± 4.3%) and even higher in the composition of chimeric κ,λ-IgG (21.5–39.0; average value: 32.0 ± 8.2%). The average content of IgG3 in the composition of antibodies with different light chains increased in the following order: κ,κ-IgG (3.7 ± 0.3%); κ,λ-IgG (6.6 ± 1.3%); and λ,λ-IgG (8.7 ± 7.0%). The maximum average content of IgG4 was found in the case of chimeric κ,λ-IgGs (8.5 ± 2.2%) compared with λ,λ-IgGs (2.6 ± 2.2%) and κ,κ-IgGs (2.1 ± 0.6%). Thus, chimeric antibodies are formed more efficiently in the case of IgG2 and, to a lesser extent, IgG4.

The subfractions of IgGs with affinity to S-protein (anti-S-IgG) and its receptor binding domain (anti-RBD-IgGs) were isolated by affinity chromatography on CNBr-activated Sepharose with immobilized recombinant S- protein and its RBD, as shown in [46]. Since healthy donors do not develop antibodies against SARS-CoV-2, these antibody subfractions were isolated only for patient groups 1–3. The content of anti-S-IgG and anti-RBD-IgGs in the blood plasma of donors who recovered from COVID-19 or were vaccinated with Sputnik V did not exceed 1.4% and 0.6%, respectively [46]; therefore, for each of the groups of patients, specific IgG subfractions were isolated from an equimolar mixture of 10 IgG preparations. First, the IgG mixture was applied to Sepharose with immobilized RBD, according to [46]. The anti-RBD-IgG fraction was eluted with an acidic buffer, while the fractions with no affinity for the sorbent were applied to Sepharose with immobilized S-protein. Elution with an acidic buffer made it possible to obtain antibodies with an affinity for S-protein domains other than RBD.

### 2.5. Catalase Activity of the IgG Subfractions

The catalase activity of each IgG preparation was evaluated as the average value of the apparent *k*cat of three measurements. κ,λ-IgG, anti-S-IgG and anti-RBD-IgG showed almost no activity. The values of apparent *k*cat for κ,κ-IgGs of the studied groups decreased in the following order (min^−1^): patients recovered from COVID-19 (group 1)—0.7 × 10^3^, vaccinated donors who have not previously had COVID-19 (group 2)—0.4 × 10^3^, patients vaccinated after the disease (group 3)—0.2 × 10^3^, and conditionally healthy donors (group 4)—0.2 × 10^3^. For λ,λ- IgGs, the apparent *k*cat decreased in the following order: conditionally healthy donors—0.5 × 10^3^, vaccinated donors who have not previously had COVID-19—0.5 × 10^3^, patients who recovered from COVID-19—0.2 × 10^3^, and donors vaccinated after disease—0.2 × 10^3^. Figure 7 shows a comparison of the catalase activity of IgG subfractions of the studied groups.

## 3. Discussion

In this work, for the first time, we evaluated the catalase activity of IgGs in the blood plasma of patients who recovered from a previous COVID-19 infection, vaccinated with Sputnik V before and after COVID-19 infection, and apparently healthy donors. In addition, the catalase activities of the κ,κ-IgGs, λ,λ-IgGs, and κ,λ-IgGs subfractions of the studied groups’ antibodies were assessed. It was shown that the ratio of IgGs of the different subclasses in the blood plasma could affect the catalase activity of the IgG preparations of the studied groups, as well as their subfractions. The apparent *k*cat values varied widely for each of the four groups. This may be due to the individual variations in the reaction of the immune system and, hence, the number of circulating antibodies and their characteristics [47]. The immune response to COVID-19 depends on the intensity and duration of exposure to the virus, the level of genetic resistance to viral diseases, sex, age, and the presence of comorbidities that can directly affect the immune system [48]. The immune response to vaccination is also personal and depends, for example, on age [49].

The blood plasma IgG from apparently healthy donors had catalase activity comparable to previously obtained data [39]. The plasma from the healthy donors was collected prior to the start of the COVID-19 pandemic, so the variability in the values and the presence of highly potent drugs are not associated with the possible disease. This is especially important because it has been estimated that approximately 80% of cases of COVID-19 are not reported because of being asymptomatic or having very mild symptoms [50].

The blood plasma IgGs of patients who recovered from COVID-19 (group 1) had the highest catalase activity; the median value of the apparent *k*cat for this group was statistically 1.8 times higher than that for the IgGs of the healthy donors (group 4). This may indicate the involvement of IgGs in antioxidant protection during SARS-CoV-2 infection and the existence of adaptive immune response mechanisms that increase the presence of antibodies with catalase activity in the blood when oxidative stress occurs. With COVID-19, the level of neutrophils in the blood increases, as well as their production of reactive oxygen species. Once in the bloodstream, ROS have a negative effect on blood components. For example, they can cause erythrocyte dysfunction in patients with severe COVID-19 [18]. Therefore, abzymes with catalase activity may play an important role in the regulation of ROS levels in the blood, especially considering that some canonical enzymes are mainly found in cells and rapidly lose their activity in the blood compared to antibodies, which are stable for 2–4 months [51]. At the same time, the relative increase in the level of IgG catalase activity in COVID-19 patients is not high compared to others, especially autoimmune and other diseases. For example, the catalase activity of IgGs in patients with schizophrenia was approximately 15.8 times higher than in healthy donors [39], and in patients with relapsing and relapsing–remitting multiple sclerosis and patients with secondary progressive multiple sclerosis, it was 2.1 and 1.7 times higher, respectively [30].

The level of IgG catalase activity of patients who were previously healthy and then vaccinated with Sputnik V (group 2) slightly exceeded the activity values for healthy donors (group 4), which may indicate that Sputnik V vaccination does not cause a significant disruption of the redox balance and immune response in individuals. At the same time, it was shown that vaccination against COVID-19 with mRNA vaccines can lead to an increase in ROS levels [52]. According to preliminary data, after healthy people’s treatment with Pfizer or Moderna vaccines who had not previously had COVID-19, ROS levels in the blood plasma were observed, which persisted until the second dose, after which ROS levels decreased [52].

The catalase activity of the antibodies differed according to the composition of their different light chains. Differences in the conformational flexibility, half-life, and tendency to change antibody specificity were observed between the κ,κ-Igs and λ,λ-Igs [53], which may affect the level of their catalase activity. In addition to monospecific antibodies, IgGs containing HL halves with one kappa and one lambda light chains (κ,λ-IgG; HκL-HλL) have been found in blood and milk [54]. These bispecific chimeric antibodies are formed by the exchange of HL fragments among different IgG molecules [54], which may cause their cross-antigen binding and, as a result, catalytic activity. The efficiency of antibodies’ HL-fragment exchange depends on the concentration of reducing agents in biological fluids. It was first shown that the reduced form of glutathione stimulates the reduction of disulfide bonds in IgG class 4 (IgG4), thereby increasing the exchange of HL fragments [55]. Later, it was shown that the halves exchange occurs among all IgG classes: IgG1, IgG2, IgG4, and IgG4 [56]. It can be assumed that oxidative stress, which occurs during COVID-19, can indirectly affect the level of mono- and bispecific IgGs in the blood and their relative catalytic activity.

The catalase activity was comparable for the κ,κ-IgGs and λ,λ-IgGs of the patients vaccinated with Sputnik V (group 2) and the donors vaccinated after COVID-19 (group 3). At the same time, in patients who recovered from COVID-19, the κ,κ-IgGs were significantly more active than the λ,λ-IgGs, and vice versa in the healthy donors.

The significantly decreased activity of the κ,λ-IgG preparations may be due to several factors. First, κ,λ-IgGs were more exposed to “acid shock” during two affinity chromatographies (Figure 7). The subfractions of κ,κ-IgG and λ,λ-IgG were obtained from KappaSelect and LambdaSelect columns using only one chromatography of IgGs. It is possible that the double exposure of κ,λ-IgGs to an acidic buffer could lead to a significant decrease in their catalase activity. In addition, κ,λ-IgG may not exhibit effective catalase activity because of their particular structural features.

It can be noted that for the κ,κ-IgG preparations of all of the studied groups, a relative increase in the percentage of IgG2 and IgG4 was observed, and for λ,λ-Ig, the levels of IgG3 and IgG1 increased. Interestingly, there is a relatively weak positive correlation between the level of catalase activity of κ,κ-Ig and λ,λ-Ig and the percentage content of IgG1 (Spearman coefficient: +0.41), but this is negative for IgG2, IgG3, and IgG4 (Spearman coefficients: −0.15, −0.10 and −0.15, respectively). In addition, there is a slight positive correlation between the level of catalase activity and the percentage content of IgG2 (Spearman coefficient: +0.32) and a negative one for IgG1, IgG4 and, in particular, IgG3 (Spearman coefficients: −0.32, −0.32, and −0.95, respectively). Perhaps this could explain the reduced catalase activity of IgG in the healthy donors, because they had a lower content of IgG2 and an increased content of IgG3 [56]. In the patients who recovered from COVID-19, there was a decrease in the level of IgG3, which may cause increased catalase activity of their antibodies. Thus, it cannot be ruled out that the results obtained may be related to differences in the structure and antigen-binding capacity between the IgG antibodies of different IgG subclasses.

The specific antibodies of COVID-19 patients are anti-S-IgGs and anti-RBD-IgGs. The levels of anti-S-IgGs and anti-RBD-IgGs are considered to be one of the main measured indicators of the level of immunity to SARS-CoV-2 [57,58]. Antibodies to the SARS-CoV-2 RBD of the S-protein inhibit the interaction of the virion with target cells, thereby being able to prevent infection [59]. These factors determined the choice of the RBD S-protein as a priority target for vaccines against COVID-19 [57]. Despite the physiological importance of anti-S-IgGs and anti- RBD-IgGs, the plasma content of both COVID-19 and Sputnik V vaccinated patients were quite low: 1.1–1.4% for anti-RBD-IgGs and 0.2–0.6% for other fragments of the S-protein [46].

Nevertheless, it was interesting to see whether the exchange of halves of anti-S-IgGs and anti-RBD-IgGs with antibodies having catalase activity occurred. The catalase activity of anti-S-IgGs and anti-RBD-IgGs in patients who recovered from COVID-19 and/or were vaccinated with Sputnik V was close to zero. Therefore, a significant exchange of anti-S-IgG and anti-RBD-IgG molecules with other antibodies during the course of COVID-19 is unlikely to occur. In the blood of healthy donors, abzymes with several different redox enzymatic functions have been found that may be involved in their protection from oxidative stress. In vivo, hydrogen peroxide is formed from ROS with superoxide dismutase. In addition to canonical superoxide dismutase, abzymes with superoxide dismutase activity can reduce oxygen from ^•^O_2_^−^ to H_2_O_2_ [60,61], while peroxidase and catalase Abs can neutralize hydrogen peroxide [42,43,44,45,46,47,48,49,50,51,52,53]. Taking together, we suggest that the specific repertoire of polyclonal human Abs can serve as an additional natural system of reactive oxygen species detoxification, and Abs can destroy hydrogen peroxide, mutagenic, toxic, and carcinogenic compounds. 

Overall, it cannot be ruled out that the immune system, in response to oxidative stress that occurs with COVID-19, can activate the production of antibodies with catalase activity.

## 4. Materials and Methods

### 4.1. Reagents

The following chemical reagents were used in this work: tris-hydroxymethylaminomethane (Tris, 111TRIS01KG) glycine (Q5778), dithiothreitol (020909), and Coomassie Brilliant Blue R-250 (821616) were from MP Biomedicals (Eschwege, Germany); KCl (1101–1.0), K_2_HPO_4_ (2981C270), KH_2_PO_4_ (1138B228), N,N,N’,N’-tetramethylethylenediamine (TEMED, L-0847–10,0), and sodium dodecyl sulfate (SDS, 0116B026) were purchased from Helicon (Moscow, Russia); acrylamide (DE790612.0500) and N,N’-methylenebisacrylamide (DE110269.0050) were from Dia-M (Moscow, Russia); Triton X-100 (Am-0694) and ammonium persulfate (PSA) were from Sigma (A3678St, Louis, MO, USA); protein molecular weight markers, PageRuler, 10–200 kDa (Thermo Fisher Scientific Baltics, Vilnius, Lithuania, 26630); hydrogen peroxide (H_2_O_2_) was from Sigma-Aldrich (St. Louis, MO, USA). An ELISA kit was used for the determination of the IgG subclass (A-8672, VECTOR-BEST, PO BOX 492, Novosibirsk, Russia). Two ELISA kits were used for the evaluation of the antibodies against the S-protein (5501 SARS-CoV-2-IgG-EIA-BEST VECTOR-BEST, PO BOX 492, Novosibirsk, Russia) and N-protein (5507 SARS-CoV-2-Ag-EIA-BEST VECTOR-BEST, PO BOX 492, Novosibirsk, Russia).

### 4.2. Donors and Patients

We used blood plasma samples from three groups of patients and one group of conditionally healthy donors: 

Group 1: 28 patients who recovered from COVID-19;

Group 2: 26 healthy people vaccinated according to the standard treatment with two doses of Sputnik V;

Group 3: 13 patients vaccinated with two doses of Sputnik V after the termination of COVID-19;

Group 4: 43 conditionally healthy donors without autoimmune pathologies and infectious diseases and not vaccinated with Sputnik V.

Blood samples from conditionally healthy donors were collected before the start of the pandemic. The blood of patients who recovered from COVID-19 and/or were vaccinated with Sputnik V was collected from October 2020 to May 2021, i.e., during the first wave of the coronavirus (Wuhan variant). The absence or presence of a history of COVID-19 was confirmed using enzyme-linked immunosorbent assays (ELISAs) of antibodies to S- and N-proteins of SARS-CoV-2, as per [46].

### 4.3. Antibody Purification and Analysis

Electrophoretically homogeneous polyclonal IgG preparations were obtained by affinity chromatography on Protein-G-Sepharose (GE Life Sciences, New York, NY, USA), similar to [43]. Plasma (1.5 mL) was centrifuged to remove possible sediment at 9000× *g*, diluted four times with TBS (50 mM Tris-HCl, pH 7.5, and 150 mM NaCl), applied to a Protein G-Sepharose column, and prebalanced with the same buffer. Proteins that did not bind to the sorbent were removed using TBS until zero absorption at 280 nm (A_280_). Then, nonspecifically adsorbed proteins and lipids were eluted additionally with TBST buffer—TBS containing 1% Triton X-100, and the column was rewashed with TBS. IgGs were specifically eluted with 0.1 M Gly-HCl buffer, pH 2.6. The resulting antibody fractions were neutralized with 1.0 M Tris-HCl buffer, pH 8.8. After isolation, the IgG preparations were dialyzed for 12 h at 4 °C against 50 volumes of 50 mM Tris-HCl buffer, pH 7.5. As shown earlier, this method makes it possible to obtain IgGs that do not contain impurities of other proteins [43].

Electrophoretic analysis of the IgG preparations was performed with SDS-PAGE according to the Laemmli method [62] under reducing (in the presence of DTT) and nonreducing (in the absence of DTT) conditions. Proteins were stained with Coomassie Blue R-250.

### 4.4. Isolation of Antibodies Containing κ and λ Light Chains

The IgG subfractions containing two kappa light chains (κ,κ-IgGs) or two lambda light chains (λ,λ-IgGs) and bispecific chimeric antibodies with λ and κ light chains (κ,λ-IgGs) were obtained with affinity chromatography of homogeneous polyclonal IgG preparations, according to the previously described method [54]. Chromatography was performed with the ÄKTA Start chromatograph (GE Healthcare) using kappa (KappaSelect, GE Life Sciences) or lambda (LambdaSelect, GE Life Sciences) light-chain-binding sorbents. The IgG preparations were applied to a KappaSelect column pre-equilibrated with TBS. To remove nonspecifically adsorbed proteins and lipids, the column was washed with TBS, then with TBST buffer, and again with TBS until zero absorption, as described above. The preparations containing κ,κ-IgGs and κ,λ-IgGs were eluted with 0.1 M Gly-HCl buffer, pH 2.6. Antibody fractions were neutralized with 1.0 M Tris-HCl buffer, pH 8.8, and then loaded onto the LambdaSelect column. Non-sorbent-bound κ,κ-IgGs were eluted with TBS. After washing the column with TBST and then TBS, bispecific κ,λ-IgGs were eluted with 0.1 M buffer, pH 2.6, neutralized, and dialyzed, as described above.

To obtain λ,λ-IgG, the polyclonal IgG preparations were applied to the LambdaSelect: λ,λ-IgGs and κ,λ-IgGs bound to the sorbent were eluted with acidic buffer, neutralized, and applied to the KappaSelect, as described above. The λ,λ-IgG preparations were eluted with TBS, while the κ,λ-IgGs were with an acidic buffer. Thus, the preparations of κ,κ-IgG, λ,λ-IgG, and κ,λ-IgG antibodies were obtained.

### 4.5. Isolation of Antibodies against S-Protein and RBD

Anti-S-IgGs and anti-RBD-IgGs were obtained using S-Sepharose and RBD-Sepharose on an Akta Start chromatograph (GE Life Sciences, New York, NY, USA) and as described in [46]. The sorbents with immobilized S-protein and RBD were prepared according to the standard protocol using CNBr Sepharose (GE Life Sciences, New York, NY, USA) and our previously published works [46,63]. An equimolar mixture of 10 IgG preparations was applied to 3 mL of RBD-Sepharose pre-equilibrated with 50 mM Tris-HCl, pH 7.5, containing 0.15 M NaCl (TBS). The fraction eluted with the acidic buffer was neutralized by adding 1/10 *v*/*v* 1.0 M Tris-HCl, pH 8.8, and then dialyzed against 20 mM Tris-HCl, pH 7.5. The IgG fraction not bound to RBD-Sepharose was applied to a 10 mL S-Sepharose column pre-equilibrated with TBS. The IgGs were eluted with a two-step gradient: 50 mM Tris-HCl, pH 7.5, containing 1.0 M NaCl and 0.1 M Gly-HCl, pH 2.6. The resulting fraction was also dialyzed against 20 mM Tris-HCl, pH 7.5.

### 4.6. Assay of the Catalase Activity

The catalase activity of the IgGs was determined spectrophotometrically, according to [45], using a Genesys 10S Bio UV/Visible spectrophotometer (Thermo Scientific, Waltham, MA, USA). The apparent *k*cat values characterizing the catalase activity of the IgGs (*k*cat = V (M/min)/(IgG) (M)) were determined from the decomposition of the hydrogen peroxide (H_2_O_2_) in the presence of the antibodies. The reaction mixture contained 30 mM H_2_O_2_, 50 mM potassium phosphate buffer, pH 7.0, and 0.03–0.1 mg/mL IgGs. The decrease in the optical density was recorded at λ = 240 nm (25 °C) for 2–5 min.

Each preparation’s activity was analyzed using pseudo-first-order reaction conditions—linear dependences of kinetic curves on antibody concentrations. The initial reaction rate was determined from the slopes of the linear sections of the kinetic curves of the dependence of the loss of H_2_O_2_ over time. The calculation took into account the value of the molar extinction coefficient of H_2_O_2_ Ε = 0.081 mM^−1^ cm^−1^ [64].

### 4.7. Statistical Analysis

Statistical data processing was carried out in OriginPro 2021 (OriginLab, Northampton, MA, USA) and STATISTICA 10 (StatSoft, Tulsa, OK, USA). The results are reported as the mean and deviation of at least two–three independent experiments for each IgG preparation. The significance of the differences (*p*) in the catalase activity among several groups was calculated using the Mann–Whitney test. To assess the correlation between the catalase activity, percentage of the IgG subclasses, and type of light chains in the preparations, Spearman’s rank correlation coefficients were calculated.

## 5. Conclusions

In this work, we analyzed the catalase activity of the IgGs of blood plasma from patients in three groups, those recovered from COVID-19, healthy people vaccinated with Sputnik V, and vaccinated patients who had recovered from COVID-19, and compared it with conditionally healthy donors. It was shown that the IgGs of patients who recovered from COVID-19 had statistically significantly higher catalase activity than those of healthy donors (1.8-fold), patients vaccinated with Sputnik V (1.4-fold), and patients vaccinated after COVID-19 (2.1-fold). An increase in the catalase activity of IgG in those who recovered from COVID-19 may indicate the important role of abzymes with oxidoreductase activities in regulating the level of reactive oxygen species. Observed differences in the catalase activity of IgG preparations with different light chain structures and the ratio of IgG subclasses could indicate the existence of immune response mechanisms that can regulate the presence of antibodies with catalase activity under conditions of vaccination and disease.

## Figures and Tables

**Figure 1 ijms-24-10081-f001:**
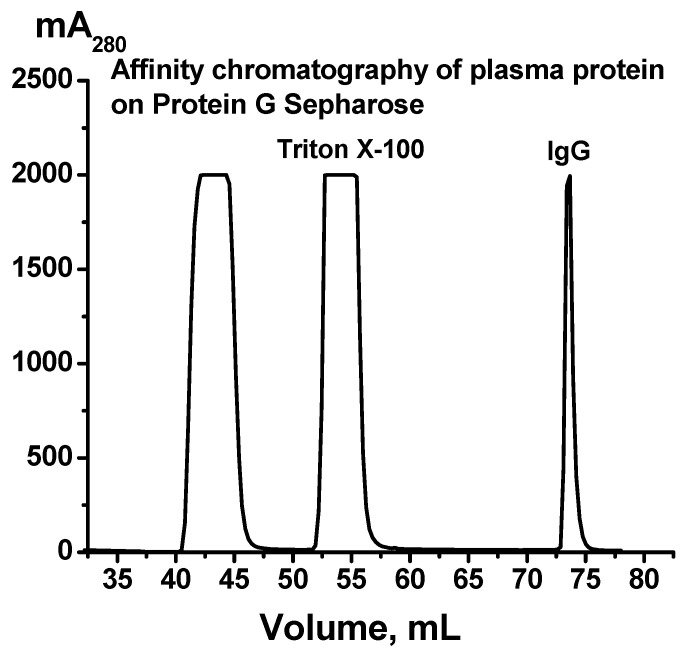
A typical profile of Protein G-Sepharose affinity chromatography of blood plasma proteins of one patient who recovered from COVID-19. The application of the plasma on the column, removal of plasma components with buffer containing 1% Triton X-100, and elution of IgGs with 50 mM Gly-HCl buffer, pH 2.6, are shown.

**Figure 2 ijms-24-10081-f002:**
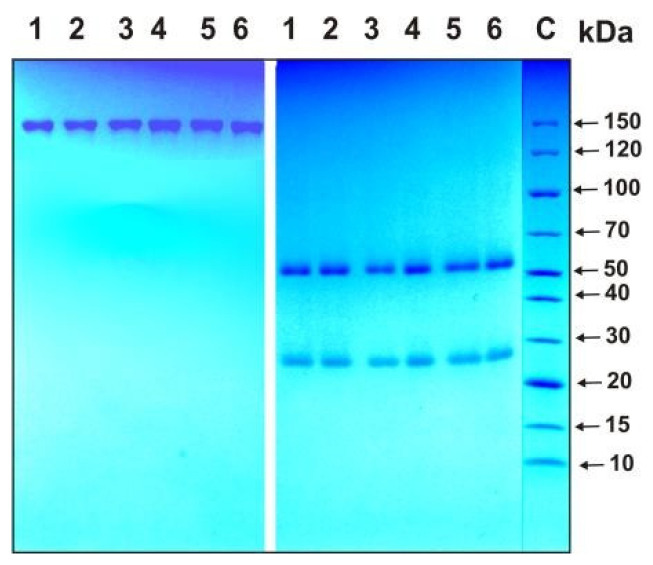
SDS-PAGE homogeneity analysis of IgG preparations before (left panel) and after treatment of IgGs with dithiothreitol (DTT; right panel): IgG preparations from 1 to 4 correspond to equimolar mixtures of antibodies (IgG_mix_) of three groups of patients (groups 1–3) and one group of healthy donors (group 4), while preparations 5 and 6 correspond to Abs of individual patients recovered from COVID-19 and healthy donors, respectively. C—protein markers with known molecular weight.

**Figure 3 ijms-24-10081-f003:**
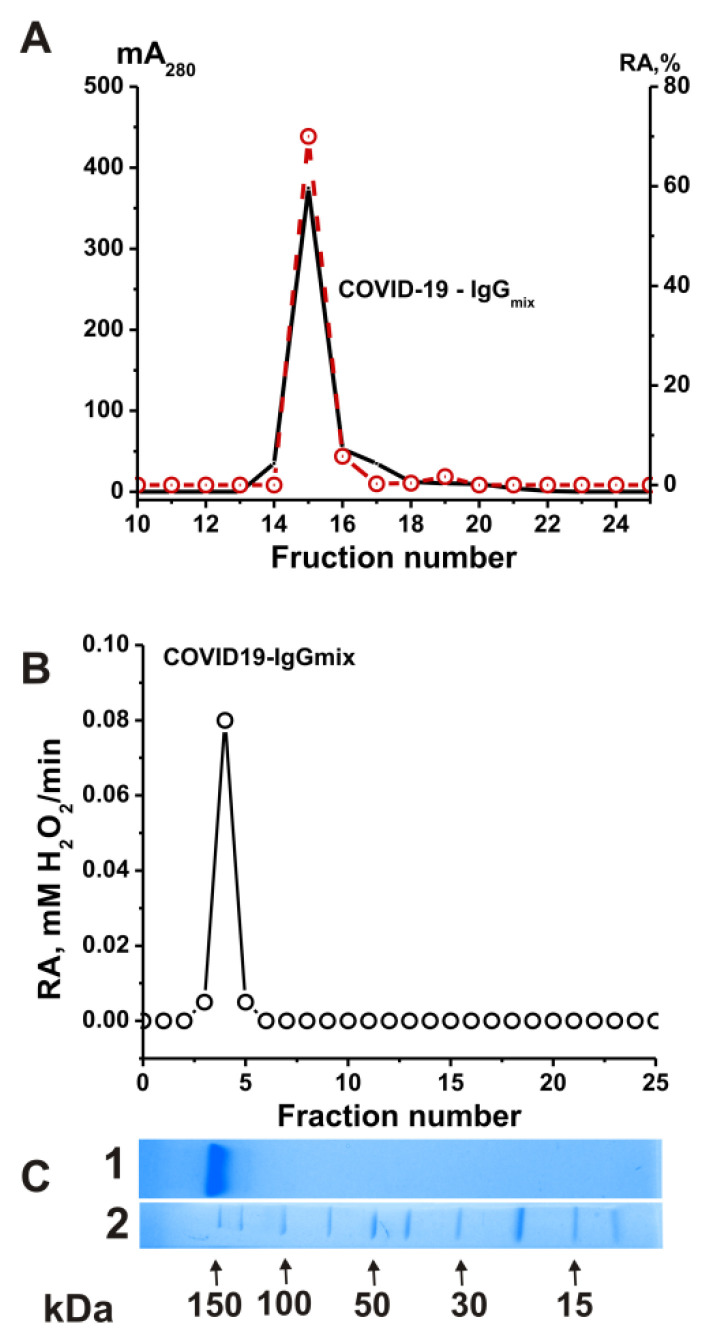
Strict criteria proved that catalase activity is an intrinsic property of the COVID-19-IgG_mix_. FPLC gel filtration of COVID-19-IgG_mix_ on a Superdex 200 column in acidic buffer (pH 2.6) after IgGs’ incubation in the same buffer: (**A**) (—), absorbance at 280 nm (A_280_); (**B**) (o), relative catalase activity (RA) of the IgG_mix_. SDS-PAGE analysis of catalase activity of the COVID-19-IgG_mix_ after SDS-PAGE in gradient 4–15 % gel. After SDS-PAGE, the gel was incubated using special conditions to remove SDS and renaturation of IgGs. The relative catalase activity (RA) was revealed (o) using the extracts of 2–3 mm gel fragments of one longitudinal slice of the gel. (**C**) In Lane 1, the second control longitudinal gel slice of the same gel was stained with Coomassie R250 to show the position of IgGs. Lane 2 (**C**) shows the position of molecular mass markers. For details, see Section 4 (Materials and Methods). The error in the initial rate determination from two independent experiments in each case did not exceed 7–10% (**A**,**B**).

**Figure 4 ijms-24-10081-f004:**
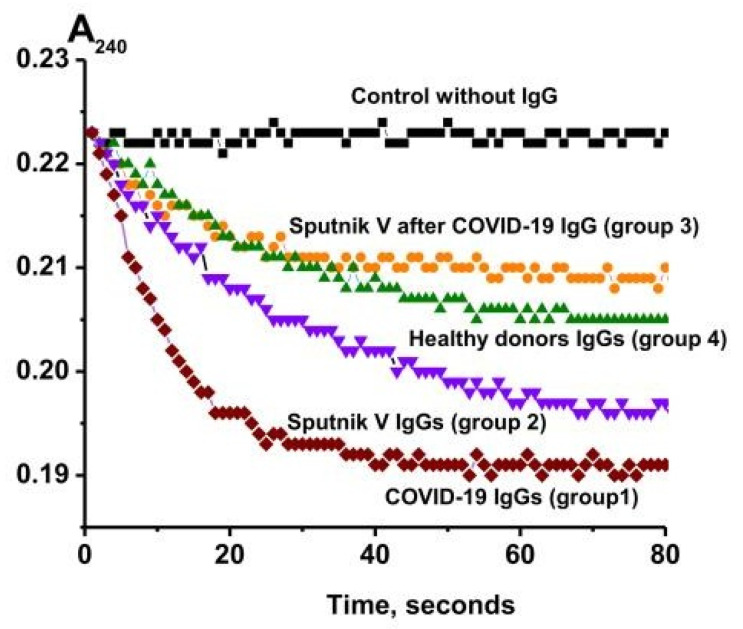
Kinetic curves of H_2_O_2_ degradation in the presence of IgGs from healthy donors (four groups): recovered from COVID-19 (group 1), healthy patients vaccinated with Sputnik V (group 2), patients vaccinated with Sputnik V after recovery (group 3), conditionally healthy donors (group 4), and a control reaction mixture containing no antibodies (control).

**Figure 5 ijms-24-10081-f005:**
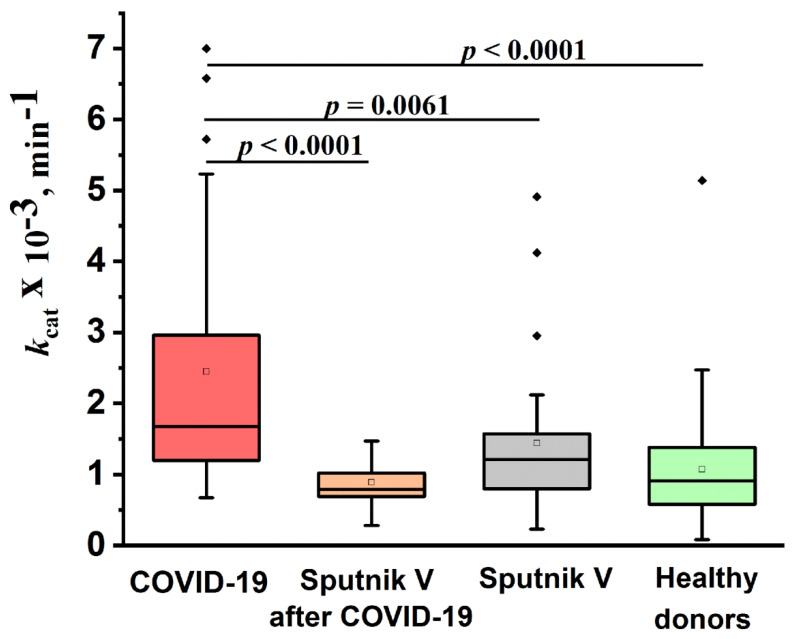
Comparison of the relative catalase activity of the IgG preparations of healthy donors (group 4), recovered from COVID-19 (group 1), healthy donors vaccinated with Sputnik V (group 2), and patients vaccinated after COVID-19 (group 3).The lower bound of the box is the first quartile of the sample (25%). The horizontal bar inside the box is the group median (50%). The upper limit of the box is the third quartile of the sample (75%). The distance between the borders of the lower and upper mustache is 1.5 of the interquartile interval. The border of the lower mustache is also the minimum of the sample. Points above the upper whisker border are outliers that fall outside the 1.5*IQ range. The empty squares inside the boxes represent the mean of the samples.

**Figure 6 ijms-24-10081-f006:**
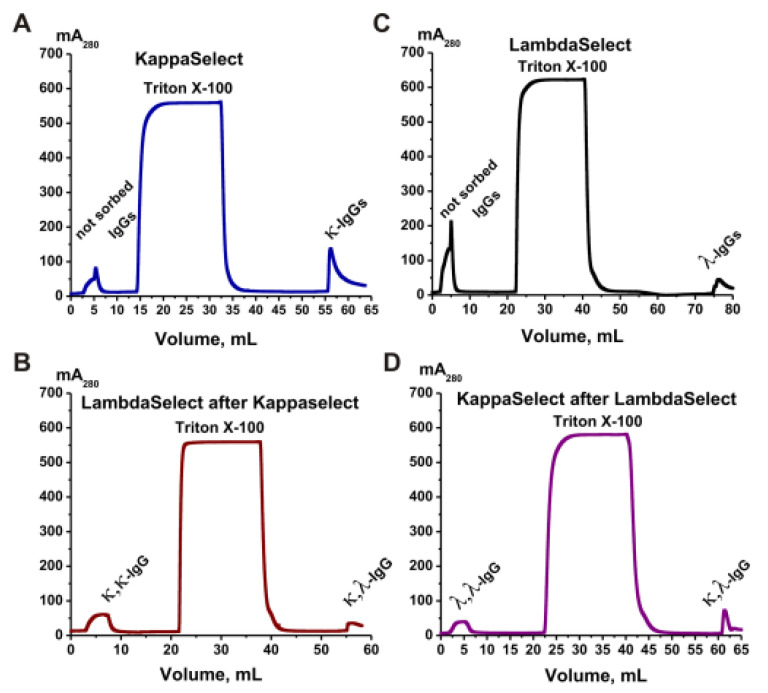
Sequential affinity chromatography of polyclonal IgGs of healthy donors on two coumns—KappaSelect and LambdaSelect (**A**,**B**) and vice versa (**C**,**D**). In the first case, κ,κ-IgGs and κ,λ-IgGs were obtained after the second chromatography (**B**). λ,λ-IgGs and κ,λ-IgGs were separated using two other chromatographies (**C**,**D**). All designations are shown in the figures.

**Figure 7 ijms-24-10081-f007:**
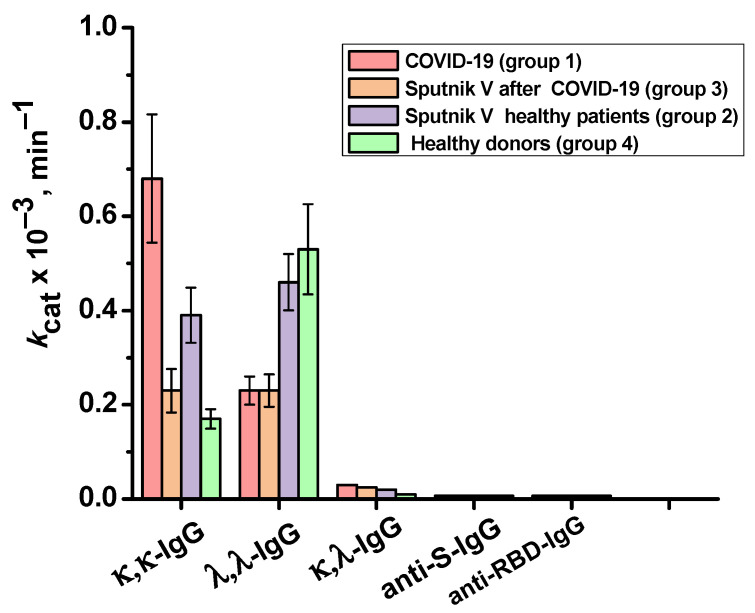
Relative catalase activity of the total preparations of κ,κ-IgG and λ,λ-IgG of healthy donors (group 4), patients who recovered from COVID-19 (group 1), patients vaccinated with Sputnik V (group 2), and patients vaccinated with Sputnik V after the disease (group 3), as well as IgGs against S-protein and BRD-protein.

**Table 1 ijms-24-10081-t001:** Characteristics of the patients and donors.

Characteristics	Group 1 (n = 28)	Group 2 (n = 26)	Group 3 (n = 13)	Group 4 (n = 42)
Males, % (n)	61% (17)	62% (16)	77% (10)	52% (22)
Age range, years (Mean ± SD)	24–72 (46.2 ± 12.4)	22–65 (42.9 ± 12.6)	19–34 (26.3 ± 5.6)	24–64 (40.5 ± 14.0)

**Table 2 ijms-24-10081-t002:** Relative content of IgG1, IgG2, IgG3, and IgG4 in the κ,κ, λ,λ, and κ,λ antibodies of the different groups *.

Type of Light Chains	Type and Number of IgG Group	Content of IgGs of the Different Subclasses, %			
		IgG1	IgG2	IgG3	IgG4
κ,κ-IgGs	Recovered from COVID-19 (group 1)	77.6	16.4	3.6	2.4
	Vaccinated with Sputnik V (group 2)	83.3	12.2	3.2	1.3
	Vaccinated after COVID-19 (group 3)	84.2	9.3	3.8	2.7
	Apparently healthy donors (group 4)	74.9	19.0	4.0	2.0
λ,λ-IgGs	Recovered from COVID-19 (group 1)	83.2	11.3	4.3	1.1
	Vaccinated with Sputnik V (group 2)	83.7	8.8	5.5	1.9
	Vaccinated after COVID-19 (group 3)	70.0	4.9	19.2	5.8
	Apparently healthy donors (group 4)	84.6	8.0	5.7	1.6
κ,λ-IgGs	Recovered from COVID-19 (group 1)	63.6	21.5	7.6	7.3
	Vaccinated with Sputnik V (group 2)	48.4	39.0	6.6	6.0
	Vaccinated after COVID-19 (group 3)	47.5	38.0	4.8	9.7
	Apparently healthy donors (group 4)	52.2	29.5	7.5	10.9

* The error in determining the relative content using data from three independent experiments did not exceed 5–10%.

## Data Availability

All data are included in the article.

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
