# Peer review of "Catalase Activity of IgGs of Patients Infected with SARS-CoV-2"

_ijms, 2023, doi:10.3390/ijms241210081_

Round 1
Reviewer 1 Report
It is a good piece of research with a huge amount of work. It is well-written, except for minor corrections. My corrections and concerns are highlighted in the PDF manuscript.

It is fine, with very few exceptions.
Author Response
It is a good piece of research with a huge amount of work. It is well-written, except for minor corrections. My corrections and concerns are highlighted in the PDF manuscript.
Answer: Thank you very much for the comments and editing the text
Best wishes
Prof Georgy A. Nevinky
Comments on the Quality of English Language
2.
Reviewer 2 Report
Manuscript is well written and has sufficient data to support hypothesis.
Author Response
It is fine, with very few exceptions.
Thank you very much for the analysis of the article
Best wishes
Prof Georgy A. Nevinky
Reviewer 3 Report
This paper describes about ”Catalase activity of IgGs of patients infected with SARS- CoV-2”, where serum IgGs have catalase activity after infection of SARS-CoV-2. It is interesting that the SARS-CoV-2 infection can lead to the hyperproduction of reactive oxygen species (ROS), which caused the possession of catalase activity of IgGs. This will contrubute to analysis the behavior of catalytic antibodies in sera of patient infected with other viruses in addition to SRAS-COV-2. If the relation between pathology and catalytic activity is revealed in future, the article must be hugely improved.
There are some comments on this paper as described in the following.
1, Table 2; Analysis the relations among kappa, lambda, IgG1, IgG2, IgG3 and IgG4 were extensively performed. Althogh there are many interesting results in the table, the authors analized them only statistical analysis were done. The relation between the structure and the ctalase activity shuld be discussed.
2, In vivo, hydrogen peroxide is produced from ROS by super oxide dismutase (SOD). The fact that IgGs have the catalase activity is directly connected with the production of hydrogen peroxide, which is synthesized by SOD. Therefore, The activity of SOD shou be stated in discussion.
3, In the paper (M. Takagi et al., FEBS letter, 375, 273-276(1995)), antibody light chain exhibited the peroxidase activity (L-zyme). Both the L-zyme and catalase have heme molecule, which strongly concerens with the decomposition of hydrogen peroxide. From the viewpoint of strucure and catalytic activity, the function of heme molecule with the IgGs exhbiting catalase activity must be taken into account in the article. For instance, the IgG uptakes the heme molecule, doesn’t it?
In addition, the kcat value of L-zyme showed 667/min. In this article, that was 200-700/min. As these kcat values are comparable, it is plausible that IgGs uptake the heme molecule.
4, In Fig.7, the order of kcat of group 1,2 3 and 4 for kk-IgG is reversed in those for ll-IgG. It seems strange. Authors should explain the reason.
NO problem
Author Response
This paper describes about ”Catalase activity of IgGs of patients infected with SARS- CoV-2”, where serum IgGs have catalase activity after infection of SARS-CoV-2. It is interesting that the SARS-CoV-2 infection can lead to the hyperproduction of reactive oxygen species (ROS), which caused the possession of catalase activity of IgGs. This will contrubute to analysis the behavior of catalytic antibodies in sera of patient infected with other viruses in addition to SRAS-COV-2. If the relation between pathology and catalytic activity is revealed in future, the article must be hugely improved.
There are some comments on this paper as described in the following.
1, Table 2; Analysis the relations among kappa, lambda, IgG1, IgG2, IgG3 and IgG4 were extensively performed. Althogh there are many interesting results in the table, the authors analized them only statistical analysis were done. The relation between the structure and the ctalase activity shuld be discussed.
Answer: Sorry, but at this time we cannot give a clear answer to this question. It is known that IgG1, IgG2, IgG3 and IgG4 differ in structure by the constant part of these antibodies as well as several other characteristics. However, data on how the difference in constant parts and other features can affect the variable regions of these antibodies is not yet available. The same applies to antibodies with kappa and lambda chains. Therefore, discussion of a possible relationship between the structure and catalase activity of antibodies may be unjustified speculation, unrelated to subtle differences in the recognition of hydrogen peroxide by these abzyme antibodies and the rate of its cleavage.
2, In vivo, hydrogen peroxide is produced from ROS by super oxide dismutase (SOD). The fact that IgGs have the catalase activity is directly connected with the production of hydrogen peroxide, which is synthesized by SOD. Therefore, The activity of SOD shou be stated in discussion.
Answer: It was done
In the blood of healthy donors, abzymes with several different redox enzymatic functions have been found that may be involved in their protection from oxidative stress. In vivo, hydrogen peroxide is formed from ROS by superoxide dismutase. In addition to canonical superoxide dismutase, abzymes with superoxide dismutase activity can reduce oxygen from ·O2– to H2O2 [60-61], while peroxidase and catalase Abs can neutralize hydrogen peroxide [42-53]. Taking together, we suggest that the specific repertoire of polyclonal human Abs can serve as an additional natural system of reactive oxygen species detoxification and Abs can destroy hydrogen peroxide, mutagenic, toxic, and carcinogenic compounds.
3, In the paper (M. Takagi et al., FEBS letter, 375, 273-276(1995)), antibody light chain exhibited the peroxidase activity (L-zyme). Both the L-zyme and catalase have heme molecule, which strongly concerens with the decomposition of hydrogen peroxide. From the viewpoint of strucure and catalytic activity, the function of heme molecule with the IgGs exhbiting catalase activity must be taken into account in the article. For instance, the IgG uptakes the heme molecule, doesn’t it?
In addition, the kcat value of L-zyme showed 667/min. In this article, that was 200-700/min. As these kcat values are comparable, it is plausible that IgGs uptake the heme molecule.
Answer: In the case of this article, M. Takagi et al obtained a recombinant antibody L chain-porphyrin Fe(III) complex. The porphyrin Fe(III) complex exhibits specific absorption in the spectrum above 280 (360-700 nM). All the natural antibodies studied by us from the blood of healthy and sick people with various catalytic activities, including catalase, demonstrated a typical spectrum of normal proteins without even any shoulders above 280 nm. In addition, when measuring catalase activity, we did not add any Fe(III) complexes. Taking this into account, from our point of view, the catalase activity of the antibodies studied by us is not associated with any of iron complexes in the preparations obtaioned. However, at the next stage of research, we will try to analyze the activity of these abzymes after the addition of the porphyrin Fe(III) complex. It is possible that the catalase activity will noticeably increase. Sorry, in this work we are not yet inclined to discuss this aspects.
4, In Fig.7, the order of kcat of group 1,2 3 and 4 for kk-IgG is reversed in those for ll-IgG. It seems strange. Authors should explain the reason.
Answer: Sorry, at this stage of research this is a very philosophical question.
In the discussion, a lot of information are given about the significant differences in the immune response of sick patients and healthy donors, as well as the structure of antibodies with lambda and kappa light chains. We cannot say for sure which of these possible many specific factors leads to a significant difference in catalase activity of kk-IgGs and ll-IgGs, and sorry we do not want to speculate.
Thank you very much for the comments
Best wishes
Prof Georgy A. Nevinky